# A Novel Scheme for Discrete and Secure LoRa Communications

**DOI:** 10.3390/s22207947

**Published:** 2022-10-18

**Authors:** Clément Demeslay, Roland Gautier, Philippe Rostaing, Gilles Burel, Anthony Fiche

**Affiliations:** CNRS UMR 6285, Lab-STICC, Universuty Brest, CNRS, CS 93837, 6 Avenue Le Gorgeu, CEDEX 3, 29238 Brest, France

**Keywords:** self-jamming waveforms, synchronization scheme, cross-correlation receiver, LoRa enhanced transceiver, LoRa discrete communications, LoRa privacy

## Abstract

In this paper, we present a new LoRa transceiver scheme to ensure discrete communications secure from potential eavesdroppers by leveraging a simple and elegant spread spectrum philosophy. The scheme modifies both preamble and payload waveforms by adapting a current state-of-the-art LoRa synchronization front-end. This scheme can also be seen as a self-jamming approach. Furthermore, we introduce a new payload demodulation method that avoids the adverse effects of the traditional cross-correlation solution that would otherwise be used. Our simulation results show that the self-jamming scheme exhibits very good symbol error rate (SER) performance with a loss of just 0.5 dB for a frequency spread factor of up to 10.

## 1. Introduction

In the past few years, LoRa has become a front-runner in low-power wide-area network (LPWAN) solutions applied to low-energy/low-cost Internet of Things (IoT) transceivers and is increasingly implemented to achieve practical solutions in areas such as agro-informatics [1], smart home design [2] and air-quality monitoring systems [3]. The increasing number of LoRa transceivers creates increased opportunities for malicious entities to disrupt or eavesdrop LoRa communications. Many studies have been conducted by the research community to evaluate the impact of jamming on performance and countermeasures have been proposed to tackle these threats. Below, we briefly review relevant studies that consider LoRa jamming schemes.

### 1.1. Previous Work on LoRa Jamming

In [4], the authors investigated the impact of traditional jammers, such as band and tone jamming, on the LoRa demodulation process and highlighted the sub-optimal energy efficiency of these jamming schemes. Other research has considered smarter and more efficient jammers involving jamming LoRa nodes with LoRa signals. In [5,6,7,8], LoRa reactive jammers (the jamming signal is only sent on detection of an incoming legitimate LoRa signal) and random jammers with a frequency hopping scheme were implemented and assessed on real-world devices. The authors concluded that jammer efficiency is obtained if the LoRa signal detection scheme is well-designed with good detection capability, and has a latency as low as possible to align the jamming signal in time with the signal of interest. In other studies, investigation of jamming where the jammer seeks to prevent a legitimate LoRa node to access the network was considered. In [9], a jammer was designed to reduce received signal strength indicator (RSSI) variations at the legitimate LoRa node, leading to an almost constantly obtained DevNonce key ID and preventing network access. The authors of [10] proposed a simple jammer detection scheme based on this philosophy, while [11,12] evaluated the jamming impact but on the global LoRa WAN network, with, for example, gateway occupancy or dropping probability metrics.

The eavesdropping case has, however, attracted less attention by the research community. To ensure secret communications, most of the proposed solutions rely on cryptographic schemes. For example, a frequency-hopping scheme was proposed in [13], while [14] introduced a reduced complexity advanced encryption system (AES) solution for the key management of LoRa WAN. Finally, recently in [15] a physical layer encryption method leveraging the randomness of the channel was presented to bypass the use of AES that imposes a burden on complexity for low-cost LoRa nodes.

### 1.2. Novelty and Contributions

In this paper, we propose a cooperative scheme between the transmitter and the receiver that further enhances [15] the scheme by improving the capacity for discrete LoRa transmission. The central notion is to leverage the well-known LoRa interference impact on demodulation but constructively by spreading the useful signal energy in the frequency space with a fixed power constraint. This can be seen as self-jamming with an added layer of spectrum spreading on top of LoRa. As the receiver is cooperative, the latter can then demodulate successfully. However, in realistic conditions, time and frequency synchronization between the transmitter and the receiver must be satisfied. We therefore propose a modified and adapted version of current state-of-the-art LoRa synchronization techniques as a solution.

The key contributions of the paper are as follows:Proposal of an enhanced scheme ensuring discrete and secure communication.A refined current LoRa synchronization front-end.Two variants of the scheme are proposed to adapt to power/complexity constraints of both uplinks and downlinks.

The remainder of the paper is organized as follows. In Section 2, we introduce the system model and some LoRa modulation basics. Section 3 presents a first approach to combatting an eavesdropper by modifying the preamble waveforms (introducing a self-jamming scheme). A modified synchronization front-end based on state-of-the-art techniques is proposed in Section 4. In Section 5, we investigate a possible threat where, in certain circumstances, an eavesdropper may synchronize itself. In Section 6, we enhance our initial self-jamming solution by proposing a modified payload demodulation scheme. Finally, we provide simulation results in Section 7 to evaluate the self-jamming method.

### 1.3. Notations

Table 1 lists the most relevant notations used throughout the paper.

## 2. System Model

### 2.1. Eavesdropping Scenario

We consider the eavesdropping scenario presented in Figure 1. There are three entities, Alice, Bob and Eve, denoted with **A**, **B** and **E** characters, respectively. **A** and **B** communicate with each other (Alice–Bob direction in the figure) in a cooperative way and exchange sensitive data that must be kept secret from eavesdroppers such as **E**. **B** has the role of the gateway and both uplink and downlink links are taken into account, depending on the **A** role. If **A** is a pure LoRa sensor, the uplink is much more critical than the downlink as the latter mainly consists of signaling traffic. However, if **A** is an actuator driven by incoming commands from **B**, for example, the downlink must be protected from **E**. We are then interested in securing both up- and downlinks and also ensuring discrete communication, reducing the intercept capability of **E**. **E** is, in this context, a fully passive receiver located sufficiently close to **A** and **B** to be able to detect both **A** or **B** LoRa signals. In this scenario, all channels separating entities are flat with additive white Gaussian noise (AWGN) and they are assumed to be symmetric. Frequency-selective channels may be considered in the future as an extension of this study.

### 2.2. LoRa Modulation Overview

LoRa waveforms are a type of chirp spread spectrum (CSS) signal. These signals rely on sine waves with instantaneous frequency (IF) that vary linearly with time over the frequency range f∈[−B/2;B/2] and the time range t∈[0;T) (*T*, the symbol period). This basic signal is called an upchirp or downchirp when IF increases or decreases with time, respectively. A Lora waveform is an *M*-ary digital modulation, comprised of *M* possible chirp modulations where the IF of the upchirp is shifted by the *M* possible values. The modulo operation is applied to ensure that the frequency remains in the interval [−B/2;B/2]. The LoRa parameters are chosen such that BT=M with $M=2^{SF}$ and SF∈{7,8,…,12} is called the spreading factor, which also corresponds to the number of bits for a LoRa symbol. In the discrete-time signal model, the chip rate ($R_{c}=1/T_{c}=M/T$) is usually used to sample the received signal, i.e., the sample period is $T_{s}=T_{c}=T/M=1/B$. The signal then has *M* samples over one symbol period *T*. Each symbol a∈{0,1,…,M−1} is mapped to an upchirp that is temporally shifted by $\tau_{a}=aT_{c}$ period. We note that a temporal shift results in a change in the initial IF.

This behavior is the heart of the *M*-ary chirp modulation. An expression of discrete LoRa waveforms sampled at $t=kT_{s}$ ($T_{s}=T_{c}$) has been derived by the authors in [16]:(1)$$x\left(kT_{s};a\right)≜x_{a}\left[k\right]=e^{2j\pi k\left(\frac{a}{M}-\frac{1}{2}+\frac{k}{2M}\right)}\;k=0,1,\ldots ,M-1.$$

The upchirp is the LoRa waveform with symbol index a=0.

### 2.3. LoRa Demodulation Scheme

The authors of [17] derived a simple and efficient solution to demodulate LoRa signals. In an AWGN flat-fading channel, the demodulation process is based on the maximum likelihood (ML) detection scheme. The received signal is:(2)$$r\left[k\right]=\alpha x_{a}\left[k\right]+w\left[k\right]$$
with $\alpha =\left|\alpha \right|e^{j\phi }$, the complex gain of the channel and w[k] an independent and identical distributed (i.i.d.) complex AWGN with zero-mean and variance $\sigma^{2}={E[|w\left[k\right]|}^{2}]$. The signal-to-noise ratio (SNR) is defined as: $SNR={\left|\alpha \right|}^{2}P_{s}/\sigma^{2}=1/\sigma^{2}$ with $P_{s}$ the transmitted signal power and, without loss of generality, we assume ${\left|\alpha \right|}^{2}=P_{s}=1$. The ML detector aims to select the frequency index *n* that maximizes the scalar product $\langle r,x_{n}\rangle \;\mathrm{for}\;n\;\in \;\left\{0,1,\ldots ,M-1\right\}$, defined as:(3)$$\begin{array}{cc} \langle r,x_{n}\rangle & =\sum_{k=0}^{M-1}r\left[k\right]x_{n}^{*}\left[k\right]\\ \; & =\sum_{k=0}^{M-1}\underset{\tilde{r}\left[k\right]}{\underbrace{r\left[k\right]x_{0}^{*}\left[k\right]}}e^{-j2\pi \frac{n}{M}k}=\tilde{R}\left[n\right] \end{array}$$

The demodulation stage proceeds with two simple operations:multiply the received waveform by a downchirp $x_{0}^{*}\left[k\right]$ (also called dechirping),compute $\tilde{R}\left[n\right]$, the discrete Fourier transform (DFT) of $\tilde{r}\left[k\right]$, and select the discrete frequency index $\hat{a}$ that maximizes $\tilde{R}\left[n\right]$.

In this way, the dechirp process merges all the signal energy into a unique frequency bin *a* that can be easily retrieved by taking the magnitude (non-coherent detection) of $\tilde{R}\left[n\right]$. The detected symbol is then:(4)$$\hat{a}=\underset{n}{arg max}\;|\tilde{R}\left[n\right]|$$

### 2.4. LoRa Frame Structure

LoRa messages are transmitted in frames that follow the specific format depicted in Figure 2.

The frame consists of a preamble followed by the payload symbols. The preamble is a critical component as it realizes the three following processes required to correctly demodulate the $N_{d}$ payload symbols:detecting the beginning of the frame by leveraging the $N_{up}$ upchirps.performing both frequency and time synchronization with the help of the $N_{up}$ upchirps and $N_{down}$ downchirps.detecting if the received frame is dedicated to the receiver by checking if the $N_{ID}=2$ consecutive network identification symbols correspond to its stored value.

LoRa transceivers generally use $N_{up}=8$, a variable $N_{d}$ value, and a fixed value $N_{down}=2.25$. The number of symbols in the preamble and the entire frame are denoted, respectively, $N_{pre}=N_{up}+N_{down}$ and $N_{frame}=N_{pre}+N_{ID}+N_{d}$.

We choose to slightly change the frame format as depicted in Figure 3 with the following modifications:Without loss of generality, the two identification symbols and the last quarter downchirp are ignored. The latter is not leveraged in the synchronization front-end. The symbol number in the frame then becomes $N_{frame}=N_{pre}+N_{d}$.We also set the condition $N_{down}=N_{up}$. This enables a balanced noise immunity between the upchirps and downchirps as these are averaged during the synchronization procedure.

The transmitted frame is then the concatenation of the upchirp, downchirp and payload symbol waveforms:(5)$$x\left[k\right]=s_{up,frame}\left[k\right]+s_{down,frame}\left[k-N_{up}M\right]+s_{data}\left[k-N_{pre}M\right]$$

## 3. Combat Basic LoRa Eavesdropper with Modified Preamble Waveform

A first approach to combat **E** is to only modify the preamble waveforms to disrupt its synchronization. A synchronization error will irredeemably lead to a demodulation error, preventing **E** from obtaining the critical data. The modified preamble waveforms are also designed to considerably increase the noise sensitivity for **E** and, thus, the discrete capacity of the scheme, while avoiding too much degradation of the performance of the link between **A** and **B**. The cooperative receiver leverages these modifications to improve its processing gain as much as possible.

The modified DFT preamble upchirp waveform in the preamble is illustrated in Figure 4. The green DFT bin depicts the legacy format. It consists of a unique DFT bin at known location $n=a_{up}=0$, containing all the signal power $M\sqrt{P_{s}}$. The basic idea of the discrete scheme is to spread the power over several DFT bins with a uniform distribution in respect of a fixed power constraint. This is represented by the DFT bins with a dashed line in the figure. The modified preamble can be written as: (6)$$\begin{array}{ccc} s_{up,frame}\left[k\right] & = & \sum_{i=0}^{N_{up}-1}s_{up}\left[k-iM\right] \end{array}$$(7)$$\begin{array}{ccc} s_{down,frame}\left[k\right] & = & \sum_{i=N_{up}}^{N_{pre}-1}s_{down}\left[k-iM\right] \end{array}$$
with: (8)$$\begin{array}{ccc} s_{up}\left[k\right] & = & \sqrt{P_{J}}\sum_{u=0}^{U-1}x_{(a_{up}-m_{up}^{u})\;\mathrm{mod}\;M}\left[k\right] \end{array}$$(9)$$\begin{array}{ccc} s_{down}\left[k\right] & = & \sqrt{P_{J}}\sum_{u=0}^{U-1}x_{(a_{down}-m_{down}^{u})\;\mathrm{mod}\;M}^{*}\left[k\right] \end{array}$$
and *U*, the number of DFT bins present, $P_{J}$, the power level of each DFT bin with $P_{J}=P_{s}/U$, $m_{up}^{u}$ and $m_{down}^{u}$, the u-th relative delay of the preamble upchirp and downchirp, respectively. We also note $m_{up}$, the associated delay vector that is sorted in ascending order, i.e., $m_{up}^{0}=0$ and $0<m_{up}^{u>0}<M$. Each $m_{up}^{u}$ delay must be unique to prevent a DFT bin overlapping issue, leading to adding DFT magnitudes and, thus, reducing the discrete capacity of the scheme. Note that U=1 and $a_{up}=0$ lead to the legacy format. The preamble downchirps follow the same structure but with $a_{down}$ and $m_{down}$ different from $a_{up}$ and $m_{up}$ to improve privacy.

Neglecting noise, the *i*-th received dechirped preamble upchirp or downchirp DFT is:(10)$$\begin{array}{ccc} {\tilde{R}}_{up}\left[n\right] & = & \alpha M\sqrt{P_{J}}\sum_{u=0}^{U-1}\delta \left[n-\left(a_{up}-m_{up}^{u}\right)\;\mathrm{mod}\;M\right] \end{array}$$(11)$$\begin{array}{ccc} {\tilde{R}}_{down}\left[n\right] & = & \alpha M\sqrt{P_{J}}\sum_{u=0}^{U-1}\delta \left[n-\left(a_{down}-m_{down}^{u}\right)\;\mathrm{mod}\;M\right] \end{array}$$

Note that each DFT bin has a null imaginary part. The DFT bin locations must remain secret from **E** to prevent its correct synchronization. $a_{up}$, $m_{up}$, $a_{down}$ and $m_{down}$ must then be random values that must be perfectly known by both **A** and **B**. That is, a specific procedure needs to be performed to satisfy this constraint. Possible solutions include the physical layer security schemes that leverage the randomness and reciprocity of the channel to enable both **A** and **B** to extract a pseudo-random bit sequence. These methods rely on the random received signal strength indicator (RSSI) variations, as LoRa transceivers have a built-in RSSI read-out feature, a solution chosen in [15], or using random channel path phase variation [18]. In practice, the **A** and **B** extracted sequences do not match perfectly and a reconciliation procedure is then necessary. This step requires the sequences exchange and may be vulnerable to eavesdroppers. The use of the Chinese remainder theorem (CRT), as in [15], or a code-word approach as in [19], are possible solutions to tackle this issue.

## 4. Self-Jamming Synchronization Front-End

In this section, we introduce desynchronizations that a receiver undergoes in practice, their effects on the LoRa demodulation, and the synchronization front-end designed to address these issues.

### 4.1. Time Desynchronization Model—Sampling Time Offset (STO)

In real conditions, the receiver continuously collects chunks of *M* samples that are not necessarily aligned with the receiver, i.e., the sampling times are different between the transmitter and the receiver. This produces a temporal window shift τ up to a symbol period *T*, as depicted in Figure 5. This effect, referred to as the sampling time offset (*STO*), introduces inter-symbol interference (ISI) if the previous symbol is different from the current symbol, i.e., $a^{-}\neq a$ and $a\neq a^{+}$ in the figure. The higher the value of τ, the greater the ISI, with maximum signal deformation when τ≈T/2.

The preamble structure prevents ISI that could degrade synchronization performance, as consecutive upchirps and downchirps are identical (see Equations (Equation 8) and (Equation 9)). τ is modeled based on the LoRa sampling frequency $F_{s}=B$ and can then be converted to a certain number of sampling periods as:(12)$$\tau =\left(\underset{STO}{\underbrace{STO_{int}+STO_{frac}}}\right)\times T_{s}$$
with $STO_{int}=\left\lfloor \tau /T_{s}\right\rceil \in \left[0;M-1\right]$, the integer number of sampling periods plus a fraction of a sampling period $STO_{frac}=STO-STO_{int}\in$[−0.5;0.5). ⌊.⌉ denotes the rounding operation to the nearest integer.

### 4.2. Frequency Desynchronization Model

Due to hardware imperfections, other desynchronizations may occur in the frequency domain, such as the carrier-frequency offset (*CFO*) and the sampling-frequency offset (*SFO*).

#### 4.2.1. Carrier-Frequency Offset (*CFO*)

As a reminder, the *CFO* is the residual carrier frequency present in the base-band signal at the receiver side. The local oscillators of the transmitter and the receiver are not perfectly centered to the desired carrier frequency $F_{c}$. A residual frequency appears, then:(13)$$\Delta_{f}=F_{c}^{t}-F_{c}^{r}$$
with $F_{c}^{t}$ (resp. $F_{c}^{r}$), the carrier frequency used by the transmitter (resp. the receiver). By analogy to the *STO*, $\Delta_{f}$ can be converted to a number of frequency bins:(14)$$\Delta_{f}=\left(\underset{CFO}{\underbrace{CFO_{int}+CFO_{frac}}}\right)\times \frac{B}{M}$$
with $CFO_{int}=\left\lfloor \Delta_{f}/\left(B/M\right)\right\rfloor \in \left[0;M-1\right]$, the integer number of DFT bins plus a fraction of a DFT bin $CFO_{frac}=CFO-CFO_{int}\in$[0;1). ⌊.⌋ denotes the floor operation.

#### 4.2.2. Sampling-Frequency Offset (*SFO*)

The *SFO* is a mismatch between the current and the desired sampling frequency at the receiver side:(15)$$F_{s}^{\prime }=F_{s}+SFO$$

In hardware implementation, and especially for low-cost IoT transceivers, such as LoRa, the same oscillator is used to perform the sampling and the carrier transposition. That is, the *CFO* and *SFO* are generated from the same source and their relationship represented as follows [20]:(16)$$SFO=\frac{B}{F_{c}}\times \Delta_{f}$$

### 4.3. Time and Frequency Desynchronization Effects on LoRa

$CFO_{int}$ and $STO_{int}$ have the effect of shifting the DFT bin position (we consider U=1 for the sake of simplicity) by a certain amount that is different when considering either upchirps: ${\hat{a}}_{up}=\left(a_{up}+\left\lfloor CFO+STO\right\rceil \right)\;\mathrm{mod}\;M$ or downchirps: ${\hat{a}}_{down}=\left(a_{down}+\left\lfloor CFO-STO\right\rceil \right)\;\mathrm{mod}\;M$. The fractional part $CFO_{frac}$ and $STO_{frac}$ progressively spread the DFT bin of interest energy to its neighbor as $CFO_{frac}$ or $STO_{frac}$ gets closer to 0.5: $n=a_{up}+1$ and $n=a_{down}-1$ for CFO; STO has the opposite behavior.

The SFO has the consequence, over time, of progressively distorting the received signal; a discrete model for LoRa is derived in [21] (considering upchirp symbols, for example, neglecting noise and channel path gains):(17)$${\tilde{r}}_{i}\left[k\right]\approx {\tilde{x}}_{a_{i}}\left[k\right]e^{2j\pi ki\left[{\left(\frac{B}{Fs^{\prime }}\right)}^{2}-\frac{B}{F_{s}^{\prime }}\right]}$$
with ${\tilde{x}}_{a_{i}}\left[k\right]$, the *i*-th received LoRa signal with symbol value $a_{i}$.

### 4.4. Synchronization Scheme

The adapted state-of-the-art LoRa synchronization front-end of our self-jamming scheme is presented in Figure 6. The front-end starts with a first pre-processing block which involves sampling the received signal at an over-sampled rate $R\times F_{s}$, dechirping $N_{up}$ blocks of *M* samples (downsampled by *R* factor), estimating and correcting $CFO_{frac}$ for these $N_{up}$ blocks, and computing the $N_{up}$ corrected DFTs. The receiver continues with the preamble detection as, in practice, the latter operates in real time.

Once the preamble is detected, the receiver re-aligns the symbols in the detected frame by $CFO_{frac}$ and estimates the other synchronization parameters, i.e., $CFO_{int}$, SFO, $STO_{int}$ and $STO_{frac}$. The estimation of both CFO and STO is not trivial. As their effects are not independent of each other, the pipeline must then be designed wisely. It finally performs a frame correction to re-align itself in time and frequency. The over-sampling by the *R* rate is required to mitigate $STO_{frac}$.

#### 4.4.1. Fractional *CFO* Correction and Preamble Detection

$CFO_{frac}$ can be estimated and compensated in this step. As the $CFO_{frac}$ estimator found in [22] has low sensitivity to the presence of multiple DFT peaks and operates blindly, we choose then to use this estimator. To ensure correct $CFO_{frac}$ estimation, no energy other than AWGN must be present in the left and right adjacent DFT bins of each of the *U* DFT peaks. We set the constraint of choosing delays with a minimal gap of ϵ DFT positions between each. This is also valid for proper $STO_{frac}$ estimation. Satisfying the constraint ϵ, the maximum number of virtual paths *U* value is:(18)$$U_{max}=\left\lfloor \frac{M}{\epsilon }-1\right\rceil$$
giving $U_{max}=25$ for ϵ=5 and SF=7, for example. In [22], the authors proposed an estimator that relies on the well-known three spectral lines (TSL) scheme by deriving ${\hat{CFO}}_{frac}$ over $N_{up}$ consecutive symbols. Each $N_{up}$ received desynchronized symbol $y_{i}\left[k\right]$ is then corrected:(19)$$y_{i}^{\prime }\left[k\right]=y_{i}\left[k\right]e^{-2j\pi k\frac{{\hat{CFO}}_{frac}}{M}}$$

The preamble detection relies on detecting the presence of consecutive demodulated symbols. With very low AWGN and a well-aligned received signal, $N_{up}$ identical and consecutive symbols should be detected but the noise progressively introduces errors and, in practice, it is very difficult to detect this specific pattern. To improve the detection performance at the cost of an increased false alarm rate, we set the constraint to detect at least *L* consecutive symbols having a maximum value difference of ±1.

Due to the presence of multiple DFT peaks of the same magnitude, the classic demodulation scheme in (Equation 4) is not suitable as the detected DFT peak location will change over the $N_{up}$ upchirps. To tackle this issue, we propose a cross-correlation approach. As the relative delays $m_{up}$ are perfectly known by the receiver, the latter can rebuild locally the expected dechirped preamble upchirp with assumed transmitted power $P_{s}=1$. This is denoted ${\tilde{S}}_{up}^{ref}\left[n\right]$. Then, for *L* consecutive received dechirped symbols, it computes the circular cross-correlation and extracts the maximum argument: (20)$$\begin{array}{ccc} F_{up,l}^{\prime }\left[m\right] & = & \sum_{n=0}^{M-1}|{\tilde{S}}_{up}^{ref}\left[n\right]||{\tilde{Y}}_{l}^{\prime }\left[\left(n-m\right)\;\mathrm{mod}\;M\right]| \end{array}$$(21)$$\begin{array}{ccc} n_{l} & = & \underset{m}{arg max}\;F_{up,l}^{\prime }\left[m\right] \end{array}$$
with p≤l≤p+(L−1), $p=\{0,1,\ldots ,p_{max}\}$, 0≤m≤M−1 and ${\tilde{Y}}_{l}^{\prime }\left[n\right]$, the DFT of ${\tilde{y}}_{l}^{\prime }\left[k\right]$. Note that $p_{max}$ is the last block of *L* demodulated symbols until preamble detection. Equation (Equation 20) can be efficiently computed with a fast Fourier transform (FFT) algorithm as:(22)$$F_{up,l}^{\prime }=IFFT\left(FFT\left(|{\tilde{S}}_{up}^{ref}|\right)\times {\left\{FFT\left(|{\tilde{Y}}_{l}^{\prime }|\right)\right\}}^{*}\right)$$

The preamble is detected if $\left(n_{p+i}+j\right)\;\mathrm{mod}\;M=n_{p}$ for i={1,2,…,L−1} and j={−1,0,1}. Once the preamble is detected, the rest of the symbols in the frame are corrected by ${\hat{CFO}}_{frac}$.

#### 4.4.2. Half Fractional *STO* Detection

As previously stated in Section 4.3, as $STO_{frac}$ gets closer to 0.5, the neighbor DFT bin energy progressively increases, leading to higher noise sensitivity. When $STO_{frac}\approx 0.5$, two DFT peaks with almost the same magnitude are present, creating detection uncertainty and preventing correct $CFO_{int}$ and $STO_{int}$ estimation. That is, $STO_{frac}$ must be mitigated before, independently from $CFO_{int}$ and $STO_{int}$. The authors in [23] proposed a solution by performing an initial $STO_{frac}$ mitigation, albeit partial, to remove this uncertainty.

We propose a different approach with a binary statistical test by detecting if $STO_{frac}\approx 0.5$. We define the hypotheses $H_{0}$, $H_{1}$ as $STO_{frac}\neq 0.5$ and $STO_{frac}=0.5$, respectively. The basic idea is to evaluate the DFT magnitude difference between the peak of interest and its neighbor bin. The less the difference, the closer to 0.5 $STO_{frac}$. Below a certain difference threshold, the receiver decides $H_{1}$, otherwise $H_{0}$. The detector is designed as follows:The $N_{up}$ preamble upchirp DFTs are averaged to reduce noise sensitivity:
(23)$$\langle {\tilde{Y}}_{up}^{\prime }\left[n\right]\rangle =\frac{1}{N_{up}}\sum_{i=0}^{N_{up}-1}{\tilde{Y}}_{i}^{\prime }\left[n\right]$$The following cyclic cross-correlation is computed and normalized:
(24)$$\begin{array}{ccc} F_{up}^{\prime }\left[m\right] & = & \sum_{n=0}^{M-1}|{\tilde{S}}_{up}^{ref}\left[n\right]||\langle {\tilde{Y}}_{up}^{\prime }\left[\left(n-m\right)\;\mathrm{mod}\;M\right]\rangle | \end{array}$$
(25)$$\begin{array}{ccc} F_{up}^{\prime }\left[m\right] & = & \frac{F_{up}^{\prime }\left[m\right]}{\max_{m}\;F_{up}^{\prime }\left[m\right]} \end{array}$$We extract the left and right neighbor DFT bin magnitudes of the maximum DFT peak and compute the criterion δ:
(26)$$n_{max}^{up}=\underset{m}{arg max}\;F_{up}^{\prime }\left[m\right]$$
(27)$$\begin{array}{cc} v^{-} & =F_{up}^{\prime }\left[\left(n_{max}^{up}-1\right)\;\mathrm{mod}\;M\right]\\ v^{+} & =F_{up}^{\prime }\left[\left(n_{max}^{up}+1\right)\;\mathrm{mod}\;M\right] \end{array}$$
(28)$$\delta =1-\max\;(v^{-},v^{+})$$$STO_{frac}\approx 0.5$ is finally detected as:
(29)$$\delta \underset{H_{1}}{\overset{H_{0}}{≷}}\lambda_{STO_{frac}\approx 0.5}$$

The frame contaminated by $STO_{frac}$ is then corrected with ${\hat{STO}}_{frac}=0.5$ (if detected) by discarding the first $R\times (M-STO_{frac})$ samples. There are then $N_{up}-1$ upchirp symbols in the preamble.

Figure 7 illustrates the evolution of averaged δ, denoted 〈δ〉, as a function of $STO_{frac}=\left\{0,0.1,\ldots ,0.9\right\}$ (R=10) for several SNR values $SNR_{dB}=\left\{-15,-12,-9,-6\right\}$, U=4 and SF=7. The delays $m_{up}$ are chosen randomly and uniformly in [0;M−1] and satisfying the gap ϵ constraint.

We can see from the figure that 〈δ〉 progressively decreases as $STO_{frac}$ gets closer to 0.5 with the minimal point reached for $STO_{frac}=0.5$. 〈δ〉 has a symmetric pattern with $STO_{frac}=0.5$. The noise has the effect of flattening the curve, reducing the contrast between $STO_{frac}$ values. The threshold $\lambda_{STO_{frac}\approx 0.5}$ must be chosen wisely. A low value will increase the non-detection probability, a situation that must be avoided as far as possible. A very high value will lead to almost constant detection; the corrected frame will then have as many as $STO_{frac}$ residuals with no $STO_{frac}\approx 0.5$ detection enabled.

In simulations, $\lambda_{STO_{frac}\approx 0.5}=0.3$ is a balanced value for the LoRa SNR range of interest $SNR_{dB}=\left\{-15,-14,\ldots ,-5\right\}$. We note that adjacent values $STO_{frac}=\left\{0.4,0.6\right\}$ are almost constantly detected as $STO_{frac}=0.5$, but the residual is ±0.1, a value that has a negligible impact on demodulation performance.

Figure 8 illustrates the histograms of δ for $STO_{frac}=\left\{0,0.1,0.2,0.3,0.4,0.5\right\}$, U=4, $SNR_{dB}=-8$ and SF=7. We note that the δ statistic follows a near-Gaussian distribution as the computed cross-correlation is a sum of Rayleigh random variables (RV). With extensive simulation results, we note that this distribution is slightly *U* dependent. Furthermore, increasing SF results in similar histograms but for lower SNRs, and the derived histogram for $STO_{frac}^{sym}=1-STO_{frac}$ is nearly the same as for $STO_{frac}$ (symmetry).

#### 4.4.3. *CFO* and *STO* Integer Estimation

The next step in the synchronization front-end is to estimate $CFO_{int}$ and $STO_{int}$. The process follows the same philosophy as so far applied to the cross-correlation approach. The receiver keeps the previously computed $n_{max}^{up}$ in (Equation 26) and performs steps (Equation 23), (Equation 24), (Equation 26) for the preamble downchirps to derive $n_{max}^{down}$. $CFO_{int}$ and $STO_{int}$ are simply derived as: (30)$$\begin{array}{cc} {\hat{CFO}}_{int} & =\left\lfloor \frac{(n_{max}^{up}+n_{max}^{down})\;\mathrm{mod}\;M}{2}\right\rfloor \end{array}$$(31)$$\begin{array}{cc} {\hat{STO}}_{int} & =(a_{up}+n_{max}^{up}-{\hat{CFO}}_{int})\;\mathrm{mod}\;M \end{array}$$

The *SFO* is simply derived as:(32)$$\hat{SFO}=\left({\hat{CFO}}_{int}+{\hat{CFO}}_{frac}\right)\times \frac{B^{2}}{M\times F_{c}}$$

As stated in [23], this synchronization scheme cannot correctly detect $CFO_{int}\geq M/4$ but, in practice, it is very unlikely to have such a high value.

#### 4.4.4. Fractional *STO* Part Estimation

The final step is to estimate $STO_{frac}$ in the case where $STO_{frac}\approx 0.5$ has not been detected earlier. The scheme is based on the TSL approach proposed in [23] but with slight modifications to be functional with our self-jamming scheme. The main steps are summarized in what follows:The averaged preamble DFT upchirps $\langle {\tilde{Y}}_{up}^{\prime }\left[n\right]\rangle$ are re-aligned by removing ${\hat{CFO}}_{int}$ and ${\hat{STO}}_{int}$ shifts. This is simply effected by performing a left circular permutation.For each of the *U* DFT peaks in $\langle {\tilde{Y}}_{up}^{\prime }\left[n\right]\rangle$, we extract its value and the left and right neighbor bins as:
(33)$$w_{c,u}=\langle {\tilde{Y}}_{up}^{\prime }\left[\left(a_{up}-m_{up}^{u}+c\right)\;\mathrm{mod}\;M\right]\rangle ,\;c\in \left\{-1,0,1\right\}$$$STO_{frac}$ is finally averaged over *U* estimates as:
(34)$${\hat{STO}}_{frac}=\frac{1}{U}\sum_{u=0}^{U-1}-ℜ\left\{\Pi_{u}\right\}$$
with:
(35)$$\Pi_{u}=\frac{e\left(-h_{u}\right)w_{1,u}-e\left(h_{u}\right)w_{-1,u}}{2\times w_{0,u}-e\left(-h_{u}\right)w_{1,u}-e\left(h_{u}\right)w_{-1,u}}$$
(36)$$h_{u}=\left({\hat{STO}}_{int}+a_{up}-m_{up}^{u}\right)\;\mathrm{mod}\;M$$
(37)$$e\left(x\right)=e^{2j\pi \frac{x}{M}}$$

## 5. EVE Blind Synchronization Threat

With this modified preamble structure, **E** cannot synchronize itself correctly without the knowledge of $a_{up}$, $a_{down}$, $m_{up}$ and $m_{down}$. The synchronization error heavily impacts the payload demodulation stage and then prevents **E** from eavesdropping. In this section, we evaluate the ability of **E** to blindly estimate synchronization parameters that would possibly threaten the sustainability of our scheme.

As previously stated, $CFO_{frac}$ can be blindly estimated by both **B** and **E**. However, **E** cannot synchronize itself if CFO is still present after $CFO_{frac}$ correction, i.e., $CFO_{int}\neq 0$. That is, **E** has the ability to blindly estimate $STO_{int}$ only if $CFO_{int}=0$. This situation may happen if **E** is a higher-end device with low hardware impairments and, thus, CFO<1.

In what follows, we present a blind method to extract $STO_{int}$. The basic idea is to leverage the fact that the STO introduces ISI only between the last upchirp and the first downchirp in the preamble. Then, **E** can use a $STO_{int}$ candidate approach by computing an energy cost for each candidate and selecting the one that minimizes the cost function. We denote each $STO_{int}$ candidate by $STO_{int}^{cand}\in \left\{0,1,\ldots ,M-1\right\}$. The blind extraction method is designed as follows:**E** generates a temporary replica of the received frame and voluntarily simulates a STO with value $STO_{int}^{cand}$ by discarding the first $R\times STO_{int}^{cand}$ samples, consequently modifying the time window process. It is denoted as $y_{cand}^{\prime }\left[k\right]$.It then dechirps, computes the DFT magnitude of the last preamble upchirp and the first preamble downchirp to derive the following quantities:
(38)$$\begin{array}{ccc} \gamma_{up}^{STO_{int}^{cand}} & = & \frac{1}{M}\sum_{n=0}^{M-1}|{\tilde{Y}}_{cand,N_{up}-2}^{\prime }\left[n\right]| \end{array}$$
(39)$$\begin{array}{ccc} \gamma_{down}^{STO_{int}^{cand}} & = & \frac{1}{M}\sum_{n=0}^{M-1}|{\tilde{Y}}_{cand,N_{up}-1}^{\prime }\left[n\right]| \end{array}$$To construct the minimum cost function point at $STO_{int}^{cand}=STO_{int}$, **E** needs to add a left circular permutation of one position to $\gamma_{up}^{STO_{int}^{cand}}$. The cost function is simply derived as:
(40)$$\begin{array}{cc} \gamma^{STO_{int}^{cand}} & =\gamma_{up}^{STO_{int}^{cand}}+\gamma_{down}^{STO_{int}^{cand}}\\ \gamma^{STO_{int}^{cand}=M-1} & =\max_{STO_{int}^{cand}}\;\gamma^{STO_{int}^{cand}} \end{array}$$$STO_{int}$ is finally estimated as:
(41)$${\hat{STO}}_{int}=\underset{STO_{int}^{cand}}{arg min}\;\gamma^{STO_{int}^{cand}}$$

This blind scheme has the drawback of being unable to correctly estimate $STO_{int}=M-1$ value, slightly increasing the $STO_{int}$ estimation error. Moreover, $STO_{frac}$ progressively increases the estimation error as it gets closer to 0.5, as highlighted in Section 7. If **E** has correctly estimated $STO_{int}$, it can easily estimate $STO_{frac}$ even without $a_{up}$ and $m_{up}$ knowledge in (Equation 36). **E** can select the DFT bins that are above a given threshold $\rho_{E}$ in $\langle {\tilde{Y}}_{up}^{\prime }\left[n\right]\rangle$ (Equation 23) with: (42)$$\rho_{E}=\lambda_{E}\times \max_{n}\;|\langle {\tilde{Y}}_{up}^{\prime }\left[n\right]\rangle |,\;\lambda_{E}\in \;]0;1]$$

The derived DFT bin positions set $A_{E}$ should correspond to $(a_{up}-m_{up})\;\mathrm{mod}\;M$ and, thus, $|A_{E}|=U$ in high SNR conditions, then enabling an identical $STO_{frac}$ estimation performance to the legitimate receiver if CFO<1. In such conditions, **E** successfully passes the synchronization front-end and can demodulate and retrieve the information in the payload.

We conclude that modification of the preamble only is necessary but not sufficient to ensure a discrete communication. A solution to tackle this more advanced **E** is then to also modify the payload waveform and is presented in the next section.

## 6. Combat Advanced LoRa Eavesdropper with Modified Payload Waveform

The payload waveform is modified with the same structure as for the preamble. This has the advantage of reducing scheme knowledge leaks, i.e., preamble symbols $a_{up}$, $a_{down}$, and delays $m_{up}$ and $m_{down}$. The modified payload waveform is then:   
(43)$$s_{data}\left[k\right]=\sum_{d=0}^{N_{d}-1}s_{data}^{\left(d\right)}\left[k-\left(N_{pre}+d\right)\times M\right]$$
with:(44)$$s_{data}^{\left(d\right)}\left[k\right]=\sqrt{P_{J}}\sum_{u=0}^{U-1}x_{(a_{data}^{\left(d\right)}-l_{d}-m_{data}^{d,u})\;\mathrm{mod}\;M}\left[k\right]$$
with $l_{d}$, a random shift (unknown by **E**) applied to the *d*-th payload symbol, $m_{data}^{d,u}$ the *u*-th relative delay of the *d*-th payload symbol $a_{data}^{\left(d\right)}$. We note $m_{data}^{\left(d\right)}$ the delay vector of the d-th payload symbol. Each $m_{data}^{\left(d\right)}$ may be different between payload symbols to improve privacy. Again, the receiver may use the same legacy cross-correlation approach to demodulate the payload symbol. However, the latter has the drawback of increasing interference peak magnitudes in (Equation 20) as *U* grows. This reduces the AWGN immunity and degrades the symbol detection performance.

We propose a modified cross-correlation implementation, denoted as mod cross-corr, that considerably mitigates this detrimental effect. Considering perfect synchronization, it consists of dechirping the received symbol $r_{data}^{\left(d\right)}\left[k\right]=s_{data}^{\left(d\right)}\left[k\right]+w\left[k\right]$ over multiple downchirp symbols instead of the unique downchirp $x_{0}^{*}\left[k\right]$:(45)$${\tilde{r}}_{data}^{\left(d\right)}\left[k\right]=\sum_{u=0}^{U-1}r_{data}^{\left(d\right)}\left[k\right]x_{(-m_{data}^{d,u}-l_{d})\;\mathrm{mod}\;M}^{*}\left[k\right]$$

The symbol is still estimated in the frequency domain:(46)$${\hat{a}}_{data}^{\left(d\right)}=\underset{n}{arg max}\;|{\tilde{R}}_{data}^{\left(d\right)}\left[n\right]|$$

To compare the legacy and the modified cross-correlation, we define the following criterion for the modified cross-correlation:(47)$$\eta_{mod\;cross-corr}=\frac{\left|{\tilde{R}}_{data}^{\left(d\right)}\left[a_{data}^{\left(d\right)}\right]\right|}{\frac{1}{M-1}\sum_{\begin{array}{c} 0\leq n\leq M-1\\ n\neq a_{data}^{\left(d\right)} \end{array}}^{}|{\tilde{R}}_{data}^{\left(d\right)}\left[n\right]|}$$
and for the legacy cross-correlation:(48)$$\eta_{cross-corr}=\frac{F_{data}^{\left(d\right)}\left[a_{data}^{\left(d\right)}\right]}{\frac{1}{M-1}\sum_{\begin{array}{c} 0\leq m\leq M-1\\ m\neq a_{data}^{\left(d\right)} \end{array}}^{}F_{data}^{\left(d\right)}\left[m\right]}$$

This represents the average magnitude difference between the DFT peak of interest and the interference peaks (AWGN plus cross-correlation peaks).

Figure 9 compares average η between the legacy and the modified cross-correlations as a function of $SNR_{dB}\in \left\{-15,-14,\ldots ,-6\right\}$ for several U={1,2,…,10}. We assume perfect synchronization and delays chosen randomly, respecting the ϵ constraint.

We can see from the figure that U=1 has a maximum and same average η between cross-corr and mod cross-corr as it is equivalent to the LoRa legacy demodulation scheme (Equation 4). It behaves as an upper limit as the higher average η, the higher the magnitude difference, and the better the performance. We also note that mod cross-corr has much lower *U* sensitivity. The loss between U=1 and U=10 is $\frac{6.475}{2.023}\approx 3.20$ for cross-corr against $\frac{6.475}{5.525}\approx 1.17$ for mod cross-corr at $SNR_{dB}=-6$. This solution is only sustainable if the *STO* has been correctly mitigated as would normally be the case when demodulating the payload. This modified cross-correlation is not suitable for synchronization parameter estimation as a candidate $STO_{int}$ approach is required (similar to the blind $STO_{int}$ estimation procedure) that gives poor synchronization performance.

Table 2 summarizes the parameters of our complete self-jamming scheme that the legitimate and eavesdropper receivers know, do not know, or must be kept secret from **E**, estimated with self-jamming scheme knowledge and blindly estimated. The symbols used in the table are described in Table 3. For conciseness, parameters which depend on others are not shown, e.g., $M=2^{SF}$.

Note that, from the table, the only parameter that is identically estimated by the legitimate receiver and the eavesdropper is $CFO_{frac}$. Furthermore, **E** can blindly estimate the *STO* and retrieve *U* under the right conditions (see Section 5). However, the critical payload parameters $m_{data}^{\left(d\right)}$ and $l_{d}$ are almost impossible to retrieve for **E** without using a brute-force approach, making proper demodulation very difficult.

## 7. Simulation Results

In this section, we present several simulation results to assess the self-jamming scheme. The following parameters are used, if not stated:SF=7, M=128$N_{up}=N_{down}=8$L=3R=10$F_{c}=868$ MHz, B=125 kHzCFO∈U[0.1;M/4−1=31]We assume that CFO<0.1 is very unlikely to happen in practice.STO∈U[0;M−1]|α|=1, ϕ∈U[0;2π]$P_{s}=1$, $P_{J}=P_{s}/U=1/U$$\lambda_{STO_{frac}\approx 0.5}=0.3$ϵ=5

### 7.1. Preamble Detection Performance

As **E** does not have $a_{up}$ and $m_{up}$ knowledge, the only possible preamble detection scheme for **E** is to compute the cross-correlation between two consecutive symbols as:    
(49)$$\begin{array}{ccc} F_{up,l,E}^{\prime }\left[m\right] & = & \sum_{n=0}^{M-1}|{\tilde{Y}}_{l}^{\prime }\left[n\right]||{\tilde{Y}}_{l+1}^{\prime }\left[\left(n-m\right)\;\mathrm{mod}\;M\right]| \end{array}$$
(50)$$\begin{array}{ccc} n_{l,E} & = & \underset{m}{arg max}\;F_{up,l,E}^{\prime }\left[m\right] \end{array}$$
with p≤l≤p+(L−1) and $p=\{0,1,\ldots ,p_{max}\}$. **E** also searches *L* consecutive symbols in $n_{l,E}$ with value difference ±1 to detect the preamble.

**A** and **B** also have the ability to use the modified cross-correlation to improve the preamble detection performance. However, as stated in Section 6, this approach does not demonstrate satisfactory performance if the STO is not mitigated. The preamble detection can only be performed in the presence of STO. That is, an $STO_{int}$ candidate approach must be leveraged with the same philosophy as the blind $STO_{int}$ estimation performed by **E** (see Section 5). To save computation resources, the candidate selection is only performed on the p-th received symbol and kept for the L−1 remaining symbols. The modified preamble detection scheme is:**A** or **B** generates a temporary replica of the received frame and voluntarily simulates an STO with value $STO_{int}^{cand}$ by discarding the first $R\times STO_{int}^{cand}$ samples, consequently modifying the time window process. It is denoted as $y_{cand}^{\prime }\left[k\right]$.It then computes the modified cross-correlation of the i-th received symbol and selects the maximum value for each $STO_{int}$ candidate as:
(51)$${\tilde{r}}_{up,l=p}^{STO_{int}^{cand}}\left[k\right]=\sum_{u=0}^{U-1}y_{cand,l=p}^{\prime }\left[k\right]x_{-m_{up}^{u}}^{*}\left[k\right]$$
(52)$$v_{max,l=p}^{STO_{int}^{cand}}=\max_{n}\;|{\tilde{R}}_{up,l=p}^{STO_{int}^{cand}}\left[n\right]|$$The candidate is selected as:
(53)$$STO_{int}^{cand,sel}=\underset{STO_{int}^{cand}}{arg max}\;v_{max,l=p}^{STO_{int}^{cand}}$$It then selects the maximum argument for each computed modified cross-correlation (p≤l≤p+(L−1)) associated with the chosen candidate:
(54)$$n_{l}=\underset{n}{arg max}\;|{\tilde{R}}_{up,l}^{STO_{int}^{cand}=STO_{int}^{cand,sel}}\left[n\right]|$$

Figure 10 presents the preamble detection performance comparison between the legitimate receiver and **E** as a function of $SNR_{dB}=\left\{-15,-14,\ldots ,0\right\}$ for several U={1,2,3,4,8,10,12} and SF=7. We also add the comparison between the legacy and the modified cross-correlation methods.

We can see from the figure that the preamble detection performance progressively decreases when *U* increases, even when using modified cross-correlation. This is because the same chosen $STO_{int}$ candidate is used for all the symbols in the block of *L* received symbols. That is, increasing *U* increases the error probability to $STO_{int}^{cand,sel}\neq STO_{int}$. This error propagates on all symbols and the probability of detecting *L* consecutive symbols with value difference ±1 then decreases.

For U≤3, the legacy and modified cross-correlation schemes have similar preamble detection performance, with a slight advantage for the modified cross-correlation method. However, for higher *U*, the modified cross-correlation scheme progressively outperforms the legacy cross-correlation scheme as *U* grows, with a performance difference of about 2 dB and a detection probability of 0.5 and U=12. Note that the modified cross-correlation performance is almost the same for U={8,10,12}.

**E** has much lower performance with a loss ≈4 dB between U=1 and U=12, with a detection probability of 0.5 and a loss ≥3 dB when compared to the legitimate receiver using the modified cross-correlation scheme, for a given *U*. **E** is much more prone to AWGN errors as the cross-correlation performed in (Equation 49) has two sources containing AWGN, while the reference upchirp in (Equation 20) is AWGN free.

### 7.2. Complexity Comparison between the Legacy and the Modified Cross-Correlation Methods

The considerably reduced *U* sensitivity of modified cross-correlation (see Section 6) is at the cost of increased complexity. The algorithms for both the legacy and the modified cross-correlation functions are provided in Algorithms 1 and 2.
**Algorithm 1:** Legacy cross-correlation algorithm **inputs:**
$r_{i}$: the i-th received symbol vector      m: the delays vector      $x_{\mathrm{ref}}$: the reference downchirp or upchirp vector      *M*: the constellation size **output**: s: the maximum peak index of the legacy cross-correlation**1** ${\tilde{R}}_{i}:=\mathrm{abs}\left(\mathrm{FFT}\left(r_{i}⊙x_{\mathrm{ref}}\right)\right)$**2** ${\tilde{S}}_{\mathrm{ref}}:=0_{M}$       %init *M*-size vector**3** ${\tilde{S}}_{\mathrm{ref}}\left[-m\;\mathrm{mod}\;M\right]:=M\sqrt{P_{J}}$**4** $F_{i}:=\mathrm{IFFT}\left({\mathrm{FFT}}^{*}\left({\tilde{S}}_{\mathrm{ref}}\right)⊙\mathrm{FFT}\left({\tilde{R}}_{i}\right)\right)$**5** **return**$s=arg max(F_{i})$

It is obvious that the legacy cross-correlation in Algorithm 1 does not depend on *U*; it then requires the same amount of operations irrespective of the *U*. However, in Algorithm 2, lines 2–4, *U* complex sums of *M* elements are required. That is, increasing *U* increases the complexity.
**Algorithm 2:** Modified cross-correlation algorithm
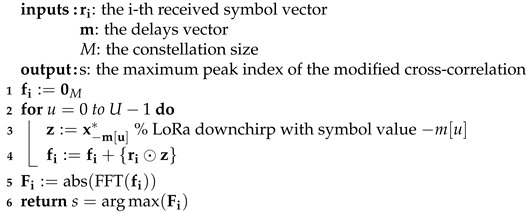


This behavior is highlighted in Figure 11. We execute and report the execution times of C compiled versions of Algorithms 1 and 2 in a MATLAB environment, with SF=7.

In Figure 11a, the mod cross-corr/legacy cross-corr execution time ratios of the preamble detection and payload demodulation processes are presented for U={1,2,…,12}. We can see for U=1 and the payload demodulation considered that mod cross-corr is about 30% faster than legacy cross-corr ($t_{exec}^{r}$
≈ 0.7). Indeed, mod cross-corr with U=1 is identical to the LoRa legacy demodulation scheme in (Equation 4). Then, computing the legacy cross-correlation for this case adds unnecessary complexity. Equally, when U=1, the $STO_{int}$ candidate procedure for preamble detection presented in Section 7.1 is useless, considerably decreasing the complexity, leading to a ratio ≈1.04. Activating the necessary $STO_{int}$ candidate approach for U>1 greatly increases the complexity cost, reflected in the high ratio transition from ≈0.7 to ≈2.8 between U=1 to U=2. Increasing *U* progressively increases the mod cross-corr complexity to reach a complexity increase factor of about 3 at U=12.

In Figure 11b, mod cross-corr and legacy cross-corr schemes are compared to the LoRa legacy demodulation when used for the payload demodulation and preamble detection processes. We note that the burden of mod cross-corr on preamble processing is much higher than that of the payload process for low *U* values but progressively reduces to reach a turnover point at U=11 where the latter increases the advantage beyond this value. Again, the $STO_{int}$ candidate approach is responsible for the high cost value at U=2 but shows less increasing complexity with *U*. The complexity of mod cross-corr is progressively increased when *U* increases to reach a factor of about 4.3 at U=12.

However, the cost of adding the legacy cross-correlation in the preamble section is very small with a constant ratio ≈1.05 as the legacy cross-correlation computation does not depend on *U*. We also note that using legacy cross-corr for the payload demodulation has higher relative complexity (≈1.45) than for the preamble detection although its absolute complexity is much lower.

Table 4 and Table 5 summarize the advantages and drawbacks of the legacy and mod cross-correlation schemes.

From Table 4, we can conclude that mod cross-corr almost completely removes *U* sensitivity and, thus, improves the frame detection and payload demodulation performances, but at the cost of increased complexity.

Table 5 shows the opposite behavior for legacy cross-corr, where it is more low-complexity compliant but has a high sensitivity with *U* which decreases the performances. That is, using mod cross-corr for the preamble detection mainly depends on performance–complexity trade-offs.

### 7.3. Integer STO Part **E** Blind Estimation Performance

Figure 12 presents the blind $STO_{int}$ estimation performance of **E** as the average estimation rate (ER) over Monte Carlo trials, defined as:
(55)$$\langle ER\rangle =\frac{1}{N_{trials}}\sum_{t=0}^{N_{trials}-1}ER\left(t\right)$$with:(56)$$ER\left(t\right)=\left\{\begin{array}{cc} 1 & \mathrm{if}\;{\hat{STO}}_{int}^{\left(t\right)}=STO_{int}^{\left(t\right)}\\ 0 & \mathrm{else} \end{array}\right.$$

The figure plots the average ER as a function of $STO_{frac}=\left\{0,0.1,\ldots ,0.9\right\}$ for random $STO_{int}\in U\left[0;M-2\right]$, fixed U=8, $CFO_{int}=0$, two $CFO_{frac}$ estimation residuals $CFO_{r}=\left\{0,0.02\right\}$ in the cases of no AWGN and several $SNR_{dB}=\left\{-3,0,3,6,9\right\}$, SF=7. We also add the legitimate receiver (**B** in the figure) performance as a comparison where the latter has the $STO_{frac}\approx 0.5$ case detection activated (see Section 4.4.2), for $SNR_{dB}=-3$ and $CFO_{frac}=0.02$.

We can see from the figure that, in a perfect $CFO_{frac}$ estimation scenario, i.e., $CFO_{r}=0$, the average ER degrades progressively as $STO_{frac}$ gets closer to 0.5. In the no AWGN case, 〈ER〉 is very good with 〈ER〉≥0.87 in the worst situation $STO_{frac}=0.5$. Increasing the noise power progressively decreases 〈ER〉 performance with 〈ER〉≤0.15 at $SNR_{dB}=-3$.

We can conclude that **E** only has synchronization capability for very high SNR environments, i.e., located very close to **A** or **B** for uplinks and downlinks, respectively. Interestingly, the $CFO_{frac}$ estimation residual produces a slightly better performance in no/very low AWGN conditions, i.e., $SNR_{dB}=\left\{\infty ,9,6\right\}$. With sufficiently low SNR, the noise finally overtakes this effect. Note that higher *U* values slightly reduce 〈ER〉 performance.

We also see that **B** has a perfect ER of 1 as the SNR value considered here is high with respect to the traditional SNR range ($SNR_{dB}<-8$ usually for SF=7) and then exhibits particularly good performance. Higher SNR values will exhibit identical performance and are not shown for the sake of figure clarity.

### 7.4. Legitimate Receiver SER Performance

Finally, we evaluate the legitimate receiver SER performance with a fully activated self-jamming scheme, i.e., modified preamble with complete synchronization and a modified cross-correlation method to demodulate payload symbols. The preamble is supposed to be detected already.

Figure 13 presents the SER performance of the legitimate receiver as a function of $SNR_{dB}=\left\{-15,-14,\ldots ,-6\right\}$ for several U={8,10,12,14,20} and SF=7. We also add the maximum performance reachable as the perfectly synchronized case with no self-jamming, i.e., U=1.

We can see from the figure that U={8,10} exhibit very good performance with a loss lower than 0.5 dB. Increasing *U* progressively degrades performance with a loss of about 3 dB for U=20. This can be explained by the fact that the legacy cross-correlation is still used in the synchronization front-end with its *U* sensitivity (see Section 6), but also because of $CFO_{frac}$ estimator limitation. If the preamble DFT peaks are too low, i.e., U≥12, $CFO_{frac}$ will not be correctly estimated in a relatively high SNR. That is, the preamble DFT averaging performed straight afterwards will not perform well; $CFO_{int}$ and $STO_{int}$ will then be incorrectly estimated, leading to a payload demodulation error. However, the U≤10 value is more than sufficient to prevent **E** from correct demodulating, as explained in the next section.

### 7.5. **E** Blind Payload Demodulation Ability

In this subsection, we investigate the ability of **E** to blindly estimate the payload symbols with the modified payload waveform scheme (see Section 6). We assume that **E** passed the synchronization front-end successfully with the advantageous but restrained conditions $SNR_{dB}\geq 6$ and CFO<1 with low $CFO_{frac}$ residual, as seen in Section 7.3. Since $m_{data}^{\left(d\right)}$ is unknown by **E**, the latter can only randomly choose one of the DFT magnitude bins that are above a given threshold $\rho_{data}^{\left(d\right)}$:(57)$$\rho_{data}^{\left(d\right)}=\lambda_{data}\times \max_{n}\;|{\tilde{S}}_{data}^{\left(d\right)}\left[n\right]|,\;\lambda_{data}\in \;]0;1]$$
with $|{\tilde{S}}_{data}^{\left(d\right)}\left[n\right]|$ the DFT magnitude of the d-th payload symbol $a_{data}^{\left(d\right)}$. The set of selected DFT bins and its length are denoted with $A_{data}$ and $\hat{U}=\left|A_{data}\right|$, respectively. For a chance for **E** to detect correctly $a_{data}^{\left(d\right)}$, the latter must be in $A_{data}$. We denote the probability that $a_{data}^{\left(d\right)}\notin A_{data}$ as $p_{A_{data}}$. This necessary condition depends on the $\lambda_{data}$ value that also drives $\hat{U}$. Then, $\lambda_{data}$ must be chosen appropriately.

Figure 14 presents the impact of $\lambda_{data}$ on average $\hat{U}$ (denoted as $\langle \hat{U}\rangle$) and $p_{A_{data}}$, respectively. We consider U=8 (a value giving very good SER performance for the legitimate receiver, as seen in Section 7.4), $SNR_{dB}=\left\{6,7,8,9\right\}$, CFO<1 with CFO estimation residual $CFO_{r}=0.02$ and random $STO_{frac}\in \left\{0,0.1,0.2,0.8,0.9\right\}$. These $STO_{frac}$ values are the range in which **E** exhibits very good $STO_{int}$ ER performance, as seen in Figure 12. In the simulation, **E** blindly estimates $STO_{int}\in \left[0;M-2\right]$ with the scheme presented in Section 5, and next performs the extraction of the DFT peaks with $\lambda_{E}$ threshold to estimate $STO_{frac}$. The estimated *STO* is compensated and **E** can finally proceed to the payload section of the frame.

From Figure 14a,b, we can see that setting $\lambda_{data}=0.1$ leads to very low $p_{A_{data}}$ as most of the DFT bins are selected, leading to a very high $\langle \hat{U}\rangle \approx 70$ at $SNR_{dB}=6$. Increasing $\lambda_{data}$ up to 0.3 decreases $\langle \hat{U}\rangle$ a great deal to reach a floor level $\langle \hat{U}\rangle \approx U=8$. Interestingly, $0.2\leq \lambda_{data}\leq 0.7$ does not impact $p_{A_{data}}$ so much with $0.02\leq p_{A_{data}}<0.1$. $\lambda_{data}>0.7$ exhibits relatively high $p_{A_{data}}$ up to ≈0.6 because of the benefit of a reduced $\langle \hat{U}\rangle \approx 4.57$ at $\lambda_{data}=0.9$ and $SNR_{dB}=6$. In this example, $\lambda_{data}=0.3$ is a good value to ensure high payload symbol capture in the DFT window of interest, i.e., $a_{data}^{\left(d\right)}\in A_{data}$ and $\langle \hat{U}\rangle \approx U$.

Nevertheless, the demodulation brute-force complexity for **E** is still prohibitively high. If we consider $\langle \hat{U}\rangle =U$, assuming that $a_{data}^{\left(d\right)}$ is always in $A_{data}$, i.e., $p_{A_{data}}=0$, and payload symbols number $N_{d}$ in the frame, this leads to the frame demodulation probability (FDP) of:(58)$$FDP=\frac{1}{U^{N_{d}}}$$

For U=8 and $N_{d}=100$, we have $U^{N_{d}}\approx 2.037\times 10^{90}$ combinations and $FDP\approx 4.909\times 10^{-90}$. At an optimistic speed of $10^{9}$ combination trials per second, this would require $6.455\times 10^{73}$ years of trials. Therefore, it prevents **E** from efficient correct demodulation.

## 8. Conclusions

In this paper, we introduced an enhanced LoRa transceiver that ensures discrete and secure communications by leveraging a simple and elegant spread spectrum philosophy. This involved first modifying the preamble LoRa waveforms to prevent eavesdropper synchronization leading to incorrect payload demodulation.

We proposed a modified synchronization scheme based on current state-of-the-art techniques that estimates and mitigates the major synchronization impairments, such as the *CFO*, *SFO* and *STO*. We added a synchronization refinement by considering the pessimistic case $STO_{frac}\approx 0.5$, previously identified in [23], and proposed an approach based on a statistical test.

We also adopted the point of view of the eavesdropper by developing a blind $STO_{int}$ estimation scheme. It exhibits good estimation performance provided that the SNR is much higher than the standard LoRa SNR range, the *CFO* is low and the received signal is well-aligned with sampling periods. Under these conditions, the eavesdropper is able to perform effective synchronization and finally retrieves the payload information. That is, modification of the preamble waveforms is necessary but not sufficient to ensure a discrete communication.

We then introduced the same modified waveform scheme to the payload but with a modified cross-correlation demodulation scheme to reduce the negative effects of the presence of multiple peaks in the LoRa DFT when using the LoRa legacy cross-correlation, at the cost of increased complexity for the legitimate receiver but much lower than that of the eavesdropper for an arbitrary small frame demodulation error. With the complete transmission scheme enabled, the SER performance loss for the legitimate receiver is less than 0.5 dB for a frequency spread factor up to U=10 at SF=7.

Table 6 summarizes the advantages and drawbacks of our LoRa self-jamming scheme. The main contribution of this scheme compared to other schemes described in the literature is the enablement of both discrete and private LoRa communications by considerably decreasing the eavesdropper’s ability to correctly identify an outgoing LoRa transmission and preventing them from proper demodulation. The potential eavesdropper will also have great difficulty in blindly synchronizing itself and collectingthe most critical system design parameters, i.e., (*U*, $m_{U}$, etc.) will only be possible with brute-force approaches. The proposed scheme is, however, not perfect and all of the advantages described are at the cost of higher implementation complexity and SER performance loss that is, however, reasonably small.

Note that this scheme does not interfere with other LoRa physical processing such as coding (e.g., Hamming and Gray coding), whitening and interleaving processes, or with the application layers, such as higher-level encryption mechanisms and LoRaWAN architecture.

From a practical implantation perspective, this scheme would require, at minimum, software modifications of existing LoRa transceivers having higher capabilities (higher computation and memory resources). This scheme may not be suitable for all applications but rather may be used for specific applications (e.g., securing a military area) where complexity constraints are not a priority but the preservation of good AWGN LoRa resilience is desired.

This analytic investigation has generated promising results for a LoRa self-jamming scheme with an adapted synchronization procedure that capitalizes on state-of-the-art LoRa synchronization algorithms. In [22], the authors evaluated the $CFO_{frac}$, $CFO_{int}$ and $STO_{int}$ estimators, as well as a variant of our $STO_{frac}$ estimator with universal software radio peripheral (USRP) equipment, and obtained good synchronization performances.

However, this scheme needs to be assessed on real-world equipment. It will be of interest to evaluate the impact of this modified waveform on the different components of the hardware front-end. For example, as this scheme adds multiple LoRa waveforms that are not necessarily coherent with each other, it may result in an increase in the peak-to-average power ratio (PAPR) and, thus, lower the performance. This may be investigated, offering interesting research opportunities for the design of modified LoRa self-jamming waveforms that can mitigate potential PAPR increase.

## Figures and Tables

**Figure 1 sensors-22-07947-f001:**
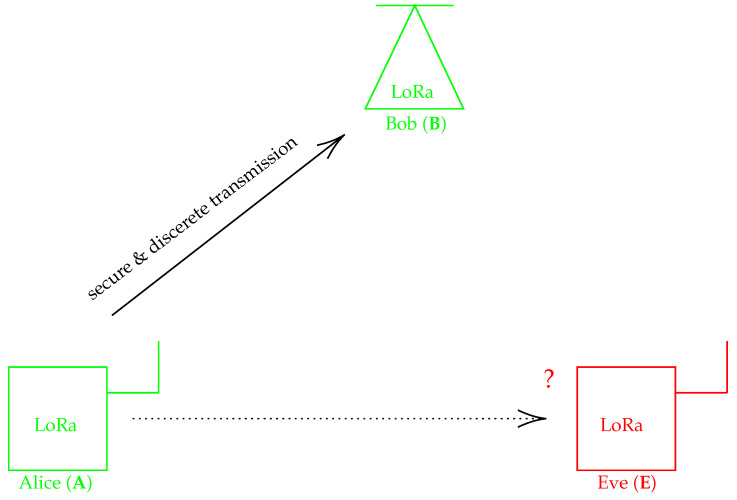
The eavesdropping scenario.

**Figure 2 sensors-22-07947-f002:**
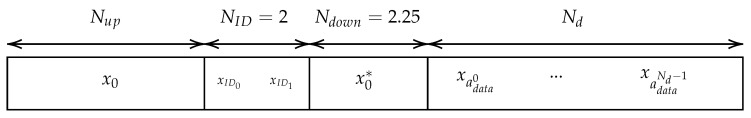
The legacy LoRa frame format.

**Figure 3 sensors-22-07947-f003:**
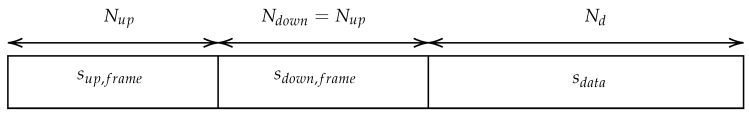
The modified self-jamming LoRa frame format.

**Figure 4 sensors-22-07947-f004:**
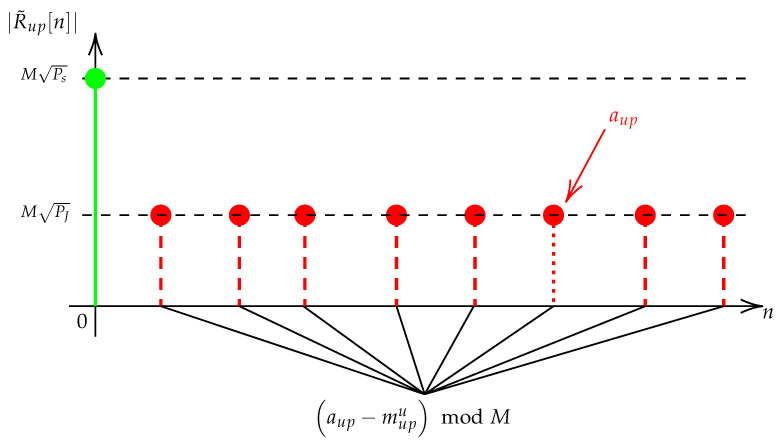
The modified preamble upchirp waveform.

**Figure 5 sensors-22-07947-f005:**
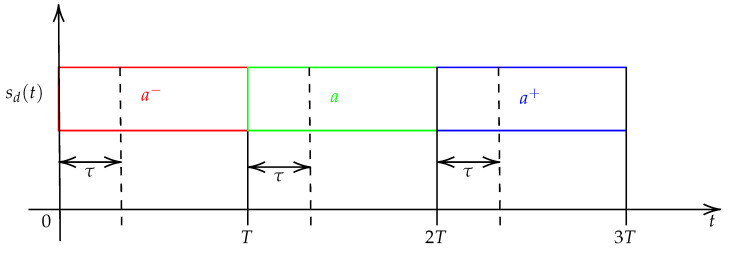
Illustration of the *STO* effect.

**Figure 6 sensors-22-07947-f006:**
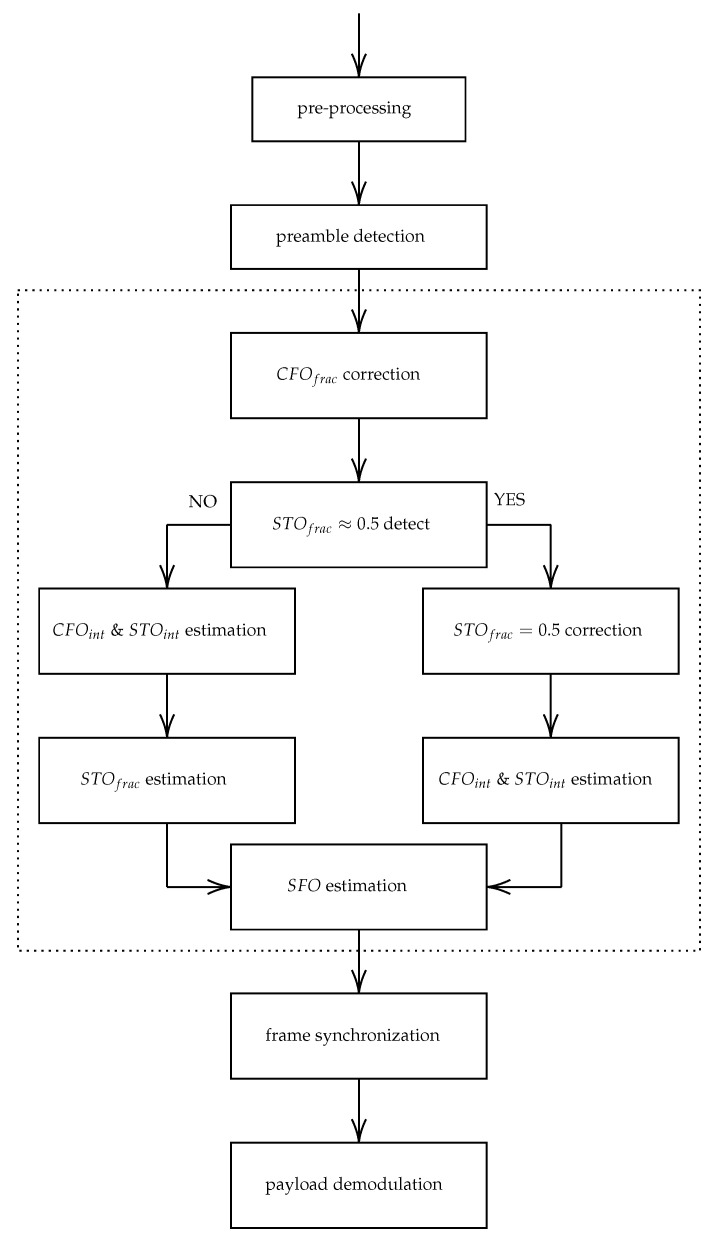
Illustration of the LoRa synchronization front-end adapted to the self-jamming scheme.

**Figure 7 sensors-22-07947-f007:**
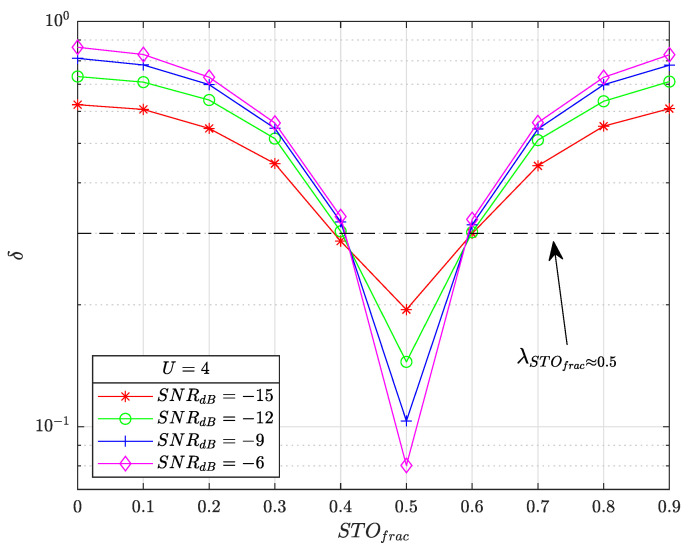
Evolution of the average value of the criterion 〈δ〉 as a function of $STO_{frac}=\left\{0,0.1,\ldots ,0.9\right\}$ for several SNR values $SNR_{dB}=\left\{-15,-12,-9,-6\right\}$, U=4 and SF=7.

**Figure 8 sensors-22-07947-f008:**
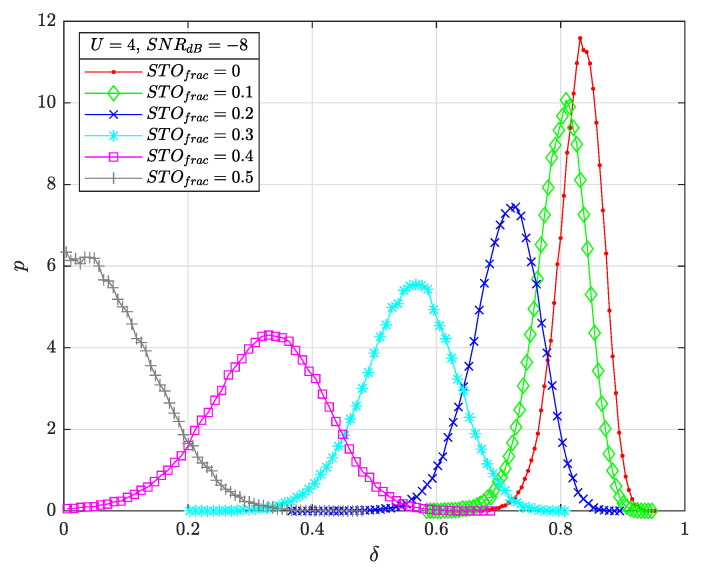
δ histograms as a function of $STO_{frac}=\left\{0,0.1,\ldots ,0.5\right\}$ for U=4, $SNR_{dB}=-8$ and SF=7.

**Figure 9 sensors-22-07947-f009:**
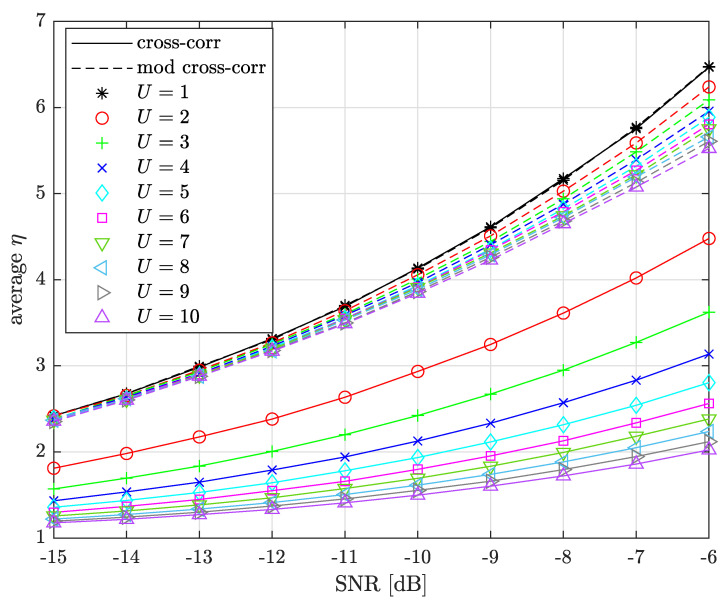
*U* sensitivity comparison between the legacy and the modified cross-correlation schemes, SF=7.

**Figure 10 sensors-22-07947-f010:**
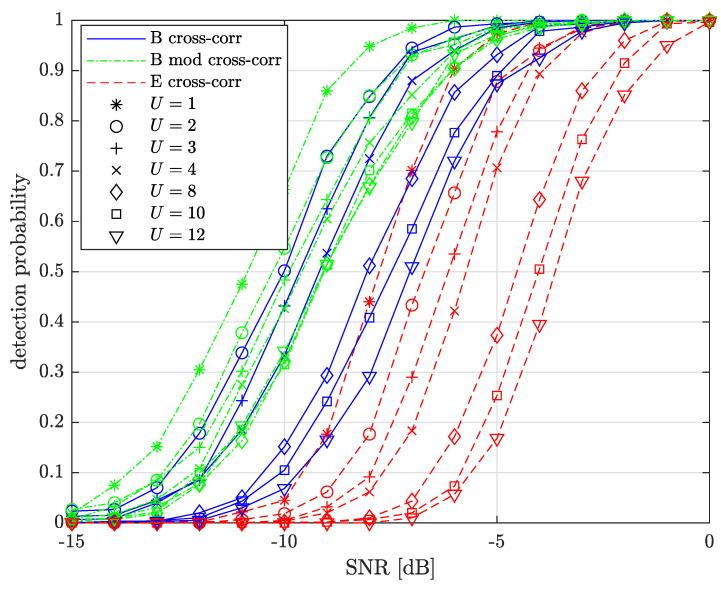
Preamble detection performance comparison between **B** and **E** for U={1,2,3,4,8,10,12}, $SNR_{dB}=\left\{-15,-14,\ldots ,0\right\}$ and SF=7. **B** can use both the legacy and the modified cross-correlation methods, while **E** is restricted to blindly detecting the preamble with the legacy cross-correlation scheme only.

**Figure 11 sensors-22-07947-f011:**
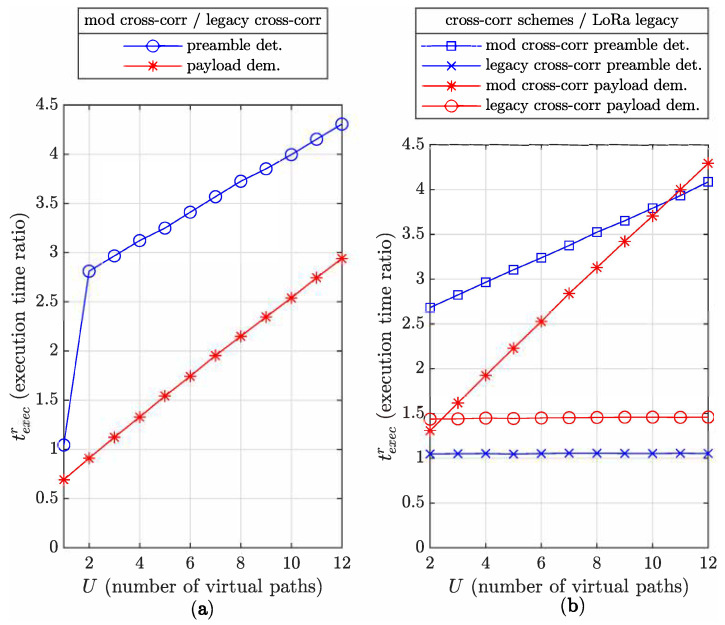
Complexity comparison for preamble detection and payload demodulation between: (**a**) mod cross-corr and legacy cross-corr. (**b**) mod cross-corr and LoRa legacy scheme, legacy cross-corr and LoRa legacy scheme.

**Figure 12 sensors-22-07947-f012:**
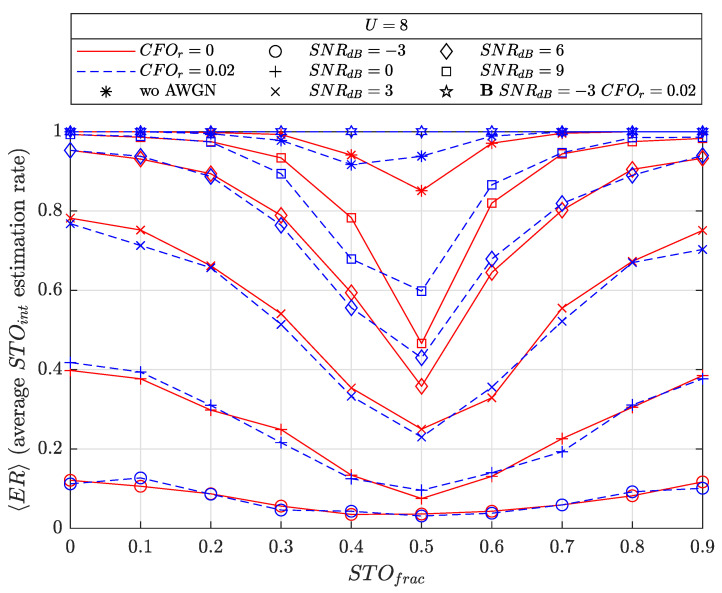
Blind $STO_{int}$ estimation performance by **E** as a function of $STO_{frac}=\left\{0,0.1,\ldots ,0.9\right\}$, U=8, no AWGN and AWGN cases with $SNR_{dB}=\left\{-3,0,3,6,9\right\}$ for the latter and SF=7. Legitimate receiver (**B**) performance is also considered for $SNR_{dB}=-3$ and $CFO_{frac}=0.02$.

**Figure 13 sensors-22-07947-f013:**
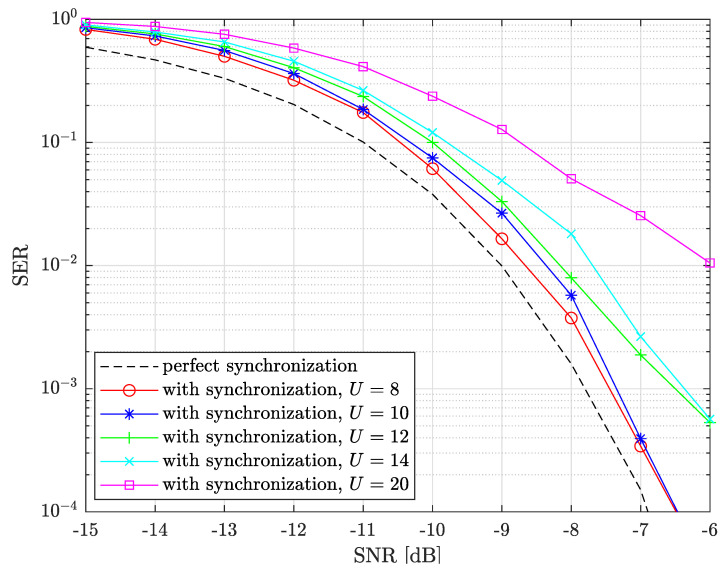
SER performance of **A** or **B** for $SNR_{dB}=\left\{-15,-14,\ldots ,-6\right\}$, several self-jamming peaks number U={8,10,12,14,20} and SF=7 with the synchronization front-end activated. The perfect synchronization case is also considered as an optimal performance bound.

**Figure 14 sensors-22-07947-f014:**
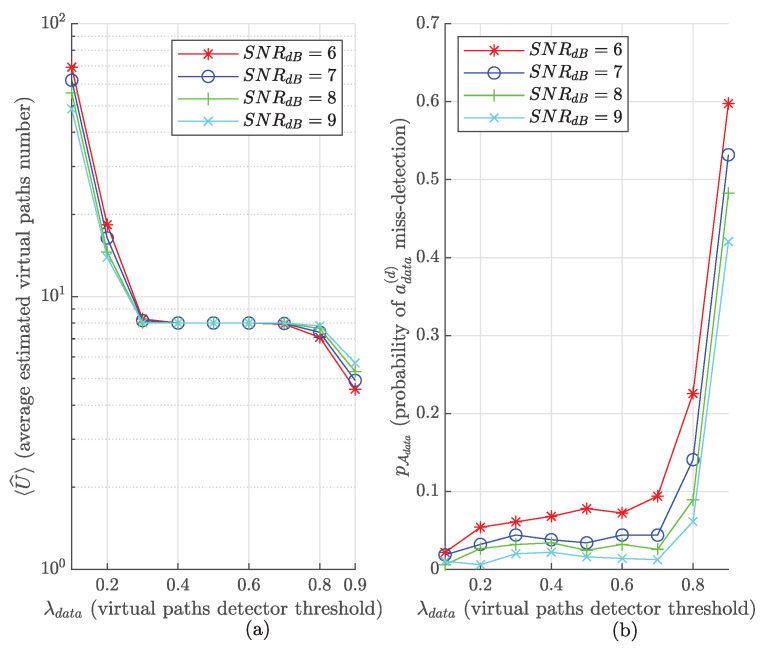
Eve blind payload demodulation performance as a function of $\lambda_{data}$ for several SNR values and SF=7. (**a**): Average estimated virtual paths number. (**b**): Probability of $a_{data}^{\left(d\right)}$ miss-detection.

**Table 1 sensors-22-07947-t001:** List of principal notations used in the paper.

Notation and Symbols Meaning	
global LoRa parameters	
SF	LoRa spreading factor
*M*	number of possible chirp waveforms per symbol: $2^{\mathrm{SF}}$
*T*	symbol period
$F_{s}$	sampling frequency
$T_{s}$	sampling period
*B*	LoRa bandwidth
$F_{c}$	carrier frequency
indexes	
*k*	time index
*n*	frequency index
*i*	symbol index
*u*	virtual path index
*m*	cross-correlation index
entities	
**A**	Alice
**B**	Bob
**E**	Eve
legacy LoRa frame parameters	
$N_{up}$	number of upchirp pilot symbols
$N_{down}$	number of downchirp pilot symbols
$N_{pre}$	number of pilot symbols: $N_{up}+N_{down}$
$N_{d}$	number of payload symbols
$N_{f}$	total number of symbols: $N_{f}=N_{pre}+N_{d}$
*a*	current transmitted symbol
$x_{a}\left[k\right]$	transmitted *a*-symbol waveform
modified LoRa frame parameters	
*U*	number of virtual channel paths
$a_{up}$	upchirp pilot symbol value
$a_{down}$	downchirp pilot symbol value
$a_{data}^{\left(d\right)}$	d-th payload symbol
$m_{up}$	vector of virtual channel delays of upchirp pilot symbols
$m_{down}$	vector of virtual channel delays of downchirp pilot symbols
ϵ	minimum DFT gap between virtual channel paths
$P_{s}$	total transmit power available
$P_{J}$	power of each virtual channel path: $P_{J}=P_{s}/U$
$S_{up}\left[k\right]$	modified upchirp preamble waveform
$S_{down}\left[k\right]$	modified downchirp preamble waveform
$S_{data}\left[k\right]$	modified data waveform
synchronization parameters	
τ	*STO* delay
$\Delta_{f}$	baseband carrier residual
$STO_{int}$, $STO_{frac}$	integer and fractional *STO* part
$CFO_{int}$, $CFO_{frac}$	integer and fractional *CFO* part
*L*	number of preamble upchirps to detect for preamble detection
${\tilde{S}}_{up}^{ref}\left[n\right]$	reference DFT upchirp for synchronization
${\tilde{S}}_{down}^{ref}\left[n\right]$	reference DFT downchirp for synchronization
$\lambda_{STO_{frac}\approx 0.5}$	threshold for $STO_{frac}\approx 0.5$ case detection
*R*	oversampling factor for $STO_{frac}$ mitigation
*various notations*	
〈x〉	averaged *x*: $\langle x\rangle =\frac{1}{N}\sum_{i=0}^{N-1}x_{i}$

**Table 2 sensors-22-07947-t002:** LoRa self-jamming scheme parameters supposed to be known, unknown, kept secret from **E**, estimated with self-jamming scheme knowledge and blindly estimated by the legitimate or eavesdropper receivers.

Self-Jamming Scheme Parameter	A or B	E
LoRa parameters		
SF	★	★
$F_{c}$, *B*	★	★
preamble waveform parameters		
$N_{up}$, $N_{down}$, $N_{d}$	★	★
$a_{up}$, $a_{down}$	★	○
$m_{up}$, $m_{down}$	★	○
payload waveform parameters		
$m_{data}^{\left(d\right)}$, $l_{d}$	★	○
$a_{data}^{\left(d\right)}$	☐	○
global self-jamming parameters		
*U*	★	△
ϵ	★	○
synchronization parameters		
*L*	★	★
$\lambda_{STO_{frac}\approx 0.5}$	★	○
$CFO_{int}$	☐	+
$CFO_{frac}$	△	△
*SFO*	☐	+
$STO_{int}$, $STO_{frac}$	☐	△

**Table 3 sensors-22-07947-t003:** Symbols meaning of symbols used in Table 2.

Symbol	Symbol Meaning
★	known
+	unknown
○	kept secret from **E**
☐	unknown and estimated with self-jamming scheme knowledge
△	unknown and blindly estimated

**Table 4 sensors-22-07947-t004:** Advantages and drawbacks of mod cross-corr.

Advantages
Mitigates *U* sensitivity
Improves frame detection performance
Improves payload demodulation performance
**Drawbacks**
Increases the complexity with *U*

**Table 5 sensors-22-07947-t005:** Advantages and drawbacks of legacy cross-corr.

Advantages
Adds low-complexity burden
Does not increase the complexity with *U*
**Drawbacks**
Leads to high sensitivity with *U*
Reduces frame-detection performance
Reduces synchronization performance

**Table 6 sensors-22-07947-t006:** Advantages and drawbacks of the LoRa self-jamming scheme.

Advantages
Enables more discrete LoRa communications
Hides sensitive information from eavesdroppers
Makes design parameter collection difficult for eavesdroppers
**Drawbacks**
Higher implementation complexity
Reasonably small SER performance loss
Software modifications required on existing LoRa transceivers

## Data Availability

Not applicable.

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
