# Peer review of "A Novel Scheme for Discrete and Secure LoRa Communications"

_sensors, 2022, doi:10.3390/s22207947_

Round 1

Reviewer 1 Report

The author has provided an enhanced LoRa transceiver with the spreading spectrum mindset. By think carefully about the possible behavior of the eavesdropper, the author proposed a cross-correlation approach for its demodulation, and build a mathematic mode for the progress. Simulation results show the chance of being wiretapped decreased a lot, which makes this work useful. But there are still some questions:

1.      The names of the variables in the article have little meaning and cause a great obstacle to reading,Readers could not follow the authors with those a,m,n,k,etc. Could author give this variables more meaningful names?

2.      You can’t use single-letter abbreviations in graph, that’s really confused. 

3.      In Figure 11, if the right part of picture is the ratio of execution time between the two method, the left one is the value of them, why is the number in left mismatch with the value in right part? Maybe there is a mistake in the axis label. And could you please add a curve of the traditional demodulation method in the graph, So that the cost for your method is more clear.

4.      Is there a mathematic symbol ]0; 1] or just a wrong way to use [ ]?

5.      In line 92, It’s better to declare the letter you use.

6.      In line 107, =2 means twice of Network IDs?

7.      Equation 5, That seems like three part of frame show up at same time, maybe there is a better way to describe this.

Author Response

Please find in the enclosed attachment our response file.

Reviewer 2 Report

The manuscript seems well written and covers an interesting and timely topic. The contribution is presented in detail and extensively analyzed, such that the results are clear. Hence, I have only a few comments as follows:

* Fig. 1 is not nice. Why is there a up- downlink drawn between A and B, but not to E? What is the use of writing cA/B = cB/A etc. on all links? It is sufficient to write that channels are assumed to be symmetric.

* It is assumed in Sec. 3 that A has a modified receiver that is capable of coping with a modified preamble waveform, while E does not have such a receiver. This assumption makes the whole scenario a bit impractical and also the analysis unfair. As a ridiculous comparison, one could also assume that E uses a WiFi transceiver, hence it is impossible to eavesdrop... just to make a point here. Is this assumption really needed? What happens if A and E have an identical receiver?

* The term "covert" seems a bit unclear in conjunction with this paper to me: From a covert communications scheme I would expect that E is unable to detect whether there is a transmission ongoing or not. However, this is not limited to applying a LoRa receiver, but also measuring, e.g., the power spectral density, or similar approaches. How does this relate to your approach?

Author Response

(The authors gave the same response as above.)

Reviewer 3 Report

The paper is a very interesting one and it may be very fruitful to extend the comments related to the proposed solution, because somehow it looks like an academic results without practical impact if we cannot implement it at large scale. An analyze based on highlighting the pros and cons aspects of the proposed solution is welcome. For example, it is unclear what is the level of compatibility of the proposed solution with the current one and how easy we may convert the old one in the new one, also for the existing equipment.

Author Response

(The authors gave the same response as above.)

Reviewer 4 Report

The authors propose a novel technique for hiding and encrypting messages sent via LoRa. LoRa makes use of a preamble to enable synchronisation between the transmitter and the receiver. The proposal is to modify the preamble so that synchronisation by an eavesdropper is made very difficult. The authors also propose similar modifications be made to the payload.

The authors claim that the advantage of the scheme is to make LoRa communication more secure.

Unfortunately, I am not persuaded that the paper is worthy of publication. Although the paper appears scientifically sound I am not convinced it makes a significant contribution. As the authors note there are other schemes for securing LoRa communication that are reasonably straightforward to implement.  It is not made clear but since this scheme is at the physical layer it would require replacement of existing LoRa hardware which is unlikely to be practical.

The paper is difficult to read. It requires a very good understanding of LoRa modulation and demodulation. The English needs considerable work to make it clearer. 

If the paper is accepted then it must make very clear what the advantages of this approach compared with other, more easily implemented approaches.

Author Response

(The authors gave the same response as above.)

Round 2

Reviewer 3 Report

The present form of the paper shows a significant improvement and it became more clear now.

Reviewer 4 Report

The paper is much improved. It is much clearer what the point is of the work they author's have done. They are not so much interested in securing LoRa as is but in identifying new LoRa like technologies with improved performance and (mainly as a side-effect) some improvement in security. I am not convinced that the paper will have a wide readership but the revised paper will be of more interest to readers than the previous version.